# Translational control by eIF2α phosphorylation regulates vulnerability to the synaptic and behavioral effects of cocaine

Wei Huang[1,2†], Andon N Placzek[1,2†‡], Gonzalo Viana Di Prisco[1,2†], Sanjeev Khatiwada[1,2,3†], Carmela Sidrauski[4,5§], Krešimir Krnjević[6], Peter Walter[4,5], John A Dani[7], Mauro Costa-Mattioli[1,2*]

[1]Department of Neuroscience, Baylor College of Medicine, Houston, United States; [2]Memory and Brain Research Center, Baylor College of Medicine, Houston, United States; [3]Verna and Marrs McLean Department of Biochemistry and Molecular Biology, Baylor College of Medicine, Houston, United States; [4]Department of Biochemistry and Biophysics, University of California, San Francisco, San Francisco, United States; [5]Howard Hughes Medical Institute, University of California, San Francisco, San Francisco, United States; [6]Department of Physiology, McGill University, Montreal, Canada; [7]Department of Neuroscience, Mahoney Institute for Neurosciences, Perelman School of Medicine, Philadelphia, United States

*For correspondence: costamat@bcm.edu

[†]These authors contributed equally to this work

Present address: [‡]Division of Basic Medical Sciences, Mercer University School of Medicine, Macon, United States; [§]Calico LLC, South San Francisco, United States

Competing interests: The authors declare that no competing interests exist.

**Abstract** Adolescents are especially prone to drug addiction, but the underlying biological basis of their increased vulnerability remains unknown. We reveal that translational control by phosphorylation of the translation initiation factor eIF2α (p-eIF2α) accounts for adolescent hypersensitivity to cocaine. In adolescent (but not adult) mice, a low dose of cocaine reduced p-eIF2α in the ventral tegmental area (VTA), potentiated synaptic inputs to VTA dopaminergic neurons, and induced drug-reinforced behavior. Like adolescents, adult mice with reduced p-eIF2α-mediated translational control were more susceptible to cocaine-induced synaptic potentiation and behavior. Conversely, like adults, adolescent mice with increased p-eIF2α became more resistant to cocaine's effects. Accordingly, metabotropic glutamate receptor-mediated long-term depression (mGluR-LTD)—whose disruption is postulated to increase vulnerability to drug addiction—was impaired in both adolescent mice and adult mice with reduced p-eIF2α mediated translation. Thus, during addiction, cocaine hijacks translational control by p-eIF2α, initiating synaptic potentiation and addiction-related behaviors. These insights may hold promise for new treatments for addiction.

## Introduction

In humans, adolescence is a period of heightened susceptibility to drug addiction (*Chambers et al., 2003*; *Kandel et al., 1992*). Although some molecular and cellular adaptations associated with drug use have been identified (*Bowers et al., 2010*; *Lüscher and Malenka, 2011*), the biological basis of heightened vulnerability to substance abuse during adolescence remains elusive. Converging evidence supports the notion that addictive drugs hijack the cellular and molecular mechanisms underlying long-term changes in synaptic strength in the mesocorticolimbic dopamine (DA) system (including the ventral tegmental area (VTA), a key brain reward area implicated in the development of addiction (*Kauer, 2004*)) in a way that reinforces drug-seeking behavior (*Bowers et al., 2010*;

**eLife digest** Drug addiction a is major mental health problem that presents a huge financial, social and legal burden worldwide. Adolescents are notoriously prone to drug abuse and addicts typically begin using drugs at a young age. However, an explanation for why young people are particularly vulnerable to the effects of addictive substances remains elusive.

Addictive drugs change how the brain works, in particular by strengthening the connections (synapses) between brain cells (neurons) and making it easier for neurons to communicate with each other. Such strengthening of synaptic connections, which can be observed when the activity of the neurons is recorded with microelectrodes, relies on new proteins being made in the brain. Since adolescents have a greater capacity than adults to make new proteins, Huang et al. hypothesized that changes in synaptic strength might occur more easily in the brain of adolescents, explaining why they are more likely to become addicted to drugs than adults.

A protein called eIF2α plays a key role in regulating the production of new proteins. Huang et al. discovered that reduced eIF2α activity accounts for why adolescents are particularly vulnerable to the synaptic and behavioral effects of cocaine. Giving adolescent mice a low dose of cocaine reduced the activity of eIF2α, caused an increase in the strength of synaptic connections in a part of the brain that processes pleasurable feelings, and promoted drug-reinforced behavior. This did not occur in adult mice.

Reducing the activity of eIF2α using either genetics or pharmacological methods caused adult mice to become as vulnerable as adolescents to cocaine-induced changes in synaptic strength and addiction-related behavior. Conversely, increasing the activity of eIF2α made adolescent mice more resistant to cocaine's effects; in other words, adolescents responded to cocaine more like adults.

Huang et al. also found that other drugs of abuse, including alcohol, methamphetamine and nicotine, all reduce eIF2α activity, suggesting that eIF2α is a common target of different drugs of abuse. In a related study, Placzek et al. investigated the role of eIF2α in nicotine addiction in mice and humans.

These findings raise several intriguing questions. How do cocaine and other drugs of abuse reduce eIF2α activity? Could variations in the activity of eIF2α or other components of the eIF2α pathway in the brain explain why some people are more likely to abuse drugs? Finally, could compounds that regulate the activity of eIF2α be useful for treating addiction?

*Kauer and Malenka, 2007*; *Hyman et al., 2006*). Addiction has both initiation and maintenance phases (*Lüscher and Malenka, 2011*; *Wise and Koob, 2014*). Here we focus on the molecular mechanisms underpinning the initial neuronal circuit adaptations caused by addictive drugs because they represent important targets for therapeutic interventions (*Lüscher and Malenka, 2011*) and are thought to contribute to the development of drug addiction (*Lüscher and Malenka, 2011*; *Kauer and Malenka, 2007*; *Kalivas et al., 2009*; *Lammel et al., 2014*). For instance, drugs of abuse (including cocaine, amphetamine, nicotine, ethanol, and morphine) *all* induce long-term potentiation (LTP) of excitatory synapses on VTA DA neurons that lasts for several days after exposure (*Bowers et al., 2010*; *Lammel et al., 2014*; *Ungless et al., 2001*; *Saal et al., 2003*). This LTP, resulting from the insertion of α-amino-3-hydroxy-5-methyl-4-isoxazolepropionic acid receptors (AMPARs) in the postsynaptic membrane, is measured by recording glutamatergic synaptic currents (EPSCs) at positive holding potentials, and is manifested as an increase in the AMPAR/*N*-methyl D-aspartate receptor (NMDAR) ratio (*Ungless et al., 2001*). Furthermore, metabotropic glutamate receptor-mediated long-term depression (mGluR-LTD), resulting from the removal of postsynaptic AMPARs, blocks cocaine-induced LTP in VTA DA neurons (*Bellone and Lüscher, 2006*). Thus, it has been postulated that impaired mGluR-LTD increases vulnerability to drugs of abuse (*Bellone and Lüscher, 2006*; *Lüscher and Huber, 2010*; *Loweth et al., 2013*).

In VTA DA neurons, protein synthesis is required for cocaine-induced LTP (*Argilli et al., 2008*; *Yuan et al., 2013*) and mGluR-LTD (*Mameli et al., 2007*). In addition, protein synthesis is also required for cocaine-induced behaviors (*Sorg and Ulibarri, 1995*; *Kuo et al., 2007*). Protein synthesis encompasses three steps: initiation, elongation, and termination. Initiation is the rate limiting

step and a major target for translational control (*Sonenberg and Hinnebusch, 2009*; *Buffington et al., 2014*). There are two main mechanisms by which translation initiation is controlled. The first is by regulation of the eIF4F complex *via* the mechanistic target of rapamycin complex 1 (mTORC1). The second mechanism is by regulating ternary complex formation *via* phosphorylation of the translation initiation factor eIF2α. Phosphorylation of eIF2α blocks general translation, but also results in translational up-regulation of a small subset of select mRNAs that contain upstream open reading frames (uORFs) in their 5' untranslated region (5'UTR) (*Sonenberg and Hinnebusch, 2009*; *Buffington et al., 2014*).

Here we report a new mechanism underlying adolescent hypersensitivity to the synaptic and behavioral effects of cocaine. In particular, we show that drugs of abuse selectively hijack the translational program controlled by phosphorylation of eIF2α in the VTA, thus potentiating synaptic inputs to VTA DA neurons and drug-induced behaviors.

## Results

### Adolescent mice are more susceptible to cocaine-evoked LTP and behavior

To examine the nature of the adolescent hypersensitivity to drugs of abuse, we first studied cocaine-induced LTP in the VTA. To this end, we recorded glutamate-mediated excitatory postsynaptic currents (EPSCs) from VTA dopaminergic (DA) neurons in midbrain slices (*Figure 1—figure supplement 1*) from adolescent (5 weeks old) and adult (3-5 months old) mice 24 hr after a single intraperitoneal (i.p.) injection of saline or cocaine (1–20 mg/kg; *Figure 1—figure supplement 2*). We used the peak amplitudes (at +40 mV) of the AMPAR and NMDAR-mediated components of the EPSCs (isolated as described (*Ungless et al., 2001*) and Methods) to calculate the AMPAR/NMDAR ratio, an index of the efficacy of synaptic transmission mediated by AMPARs. In adolescent ) mice, a relatively low dose of cocaine (5 mg/kg i.p.) elicited LTP, manifested by an increase in the AMPAR/NMDAR ratio (*Figure 1a* and *Figure 1—figure supplement 2*). By contrast, only higher doses of cocaine (10 and 20 mg/kg) induced LTP in VTA DA neurons from adult mice (*Figure 1b* and *Figure 1—figure supplement 2*). Thus, cocaine-induced LTP in VTA DA neurons is facilitated in slices from adolescent mice.

To examine whether the cocaine-induced LTP was linked to drug-related behavior, we performed conditioned place preference (CPP) tests in adolescent and adult mice. In this task, mice were first presented with either cocaine or saline in different environments. The amount of time spent in the environment previously associated with cocaine versus saline was subsequently recorded. Strikingly, we found that enhanced LTP in the VTA was mirrored in the behavior of adolescent mice: low doses of cocaine (5 mg/kg) elicited CPP only in adolescents, but not in adult mice (*Figure 1c*). Further mirroring the LTP results, higher doses of cocaine (10 mg/kg) were required to induce CPP in adult mice (*Figure 1d*). Taken together, these data indicate that adolescent mice are more sensitive to the effects of cocaine with regard to both synaptic transmission and behavior.

To further support these findings and rule out potential differences in cocaine metabolism between age groups, we applied cocaine in vitro to midbrain slices from adult and adolescent mice and conducted whole-cell recordings (*Figure 1—figure supplement 3a*), as previously described (*Argilli et al., 2008*). In slices from adolescent mice, a relatively low concentration of cocaine (1 μM) elicited LTP in VTA DA neurons (*Figure 1—figure supplement 3b*), whereas in slices from adult mice only a higher concentration of cocaine (5 μM) induced LTP (*Figure 1—figure supplement 3c*). Taken together with previous reports that the concentrations of cocaine in both blood and brain are similar in adolescent and adult mice (*Zombeck et al., 2009*), these data support the notion that cocaine-induced LTP in VTA DA neurons has a lower threshold in adolescent mice.

Since mGluR-LTD blocks cocaine-induced LTP in VTA DA neurons (*Bellone and Lüscher, 2006*) and its disruption has been postulated to enhance vulnerability to drug addiction (*Lüscher and Huber, 2010*), we predicted that mGluR-LTD in the VTA would be impaired in adolescent mice. In agreement with this prediction, a brief-application of DHPG—a selective mGluR1/5 agonist—induced LTD in VTA DA neurons from adult (*Figure 1e*), but not adolescent mice (*Figure 1e*).

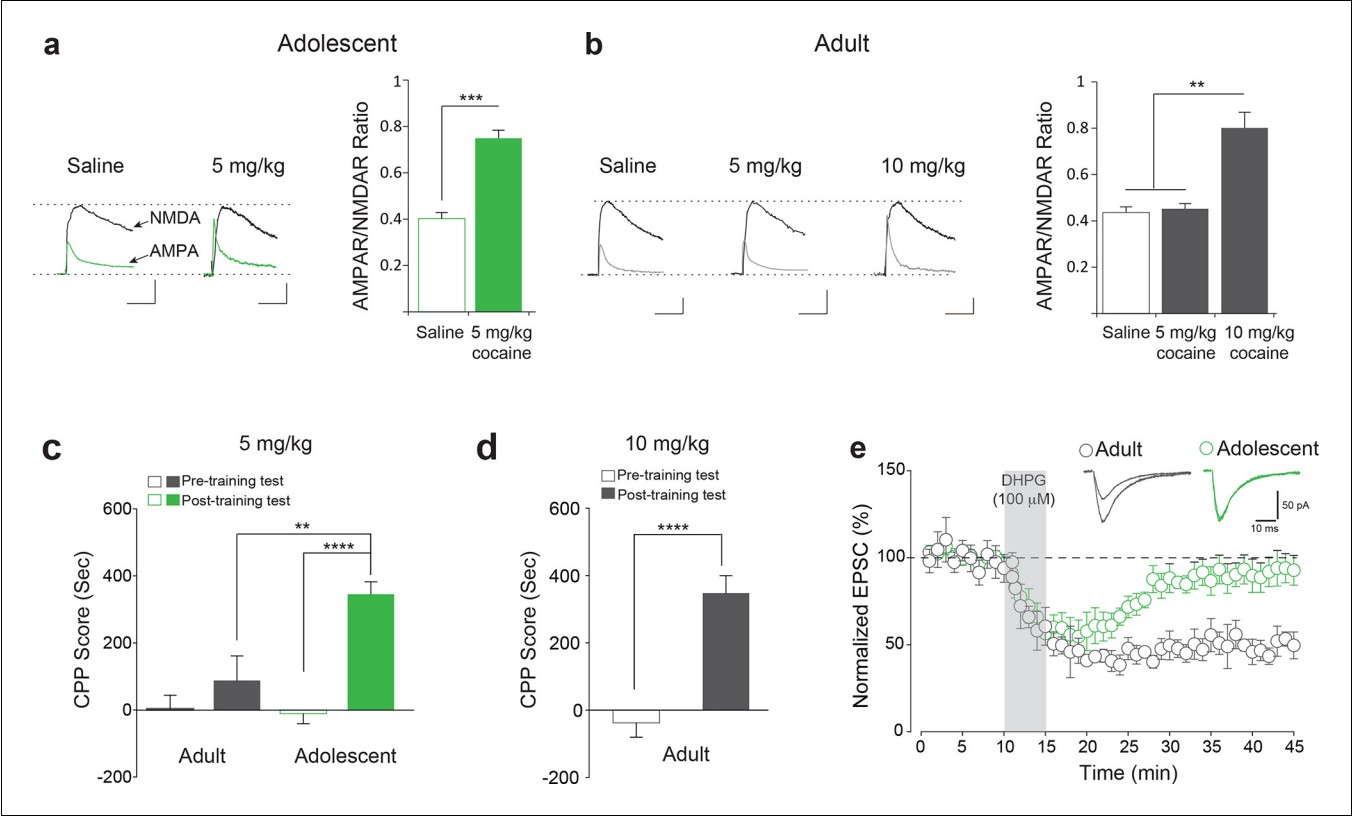

**Figure 1.** Enhanced susceptibility of adolescent mice to cocaine-induced synaptic potentiation and behavior. (**a–b**) Left, Representative traces of AMPAR and NMDAR EPSCs recorded in VTA DA neurons 24 hr after a single i.p. injection of saline or cocaine. A low dose of cocaine (5 mg/kg) induced LTP, as determined by the increase in the AMPAR/NMDAR ratio (**a**, Right, $p<0.001$, n=11/10 saline/cocaine, $t_{19}=8.09$) as well as CPP (**c**, $p<0.0001$, n=11, $t_{20}=7.487$) in adolescent mice (5 weeks old), but not in adult mice (3–5 months old, **b**, Right, $p=0.951$, n=8/9/7 saline/5 mg/kg cocaine/10 mg/kg cocaine, $F_{2,22}=27.20$; **c**, $p=0.3289$, n=9, $t_{16}=1.007$). A higher dose of cocaine (10 mg/kg) induced LTP in VTA DA neurons (**b**, Right, $p<0.01$ vs. saline or 5 mg/kg cocaine, n=8/9/7 saline/5 mg/kg cocaine/10 mg/kg cocaine, $F_{2,22}=27.20$) and CPP in adult mice (**d**, $p<0.0001$, n=15, $t_{28}=5.750$). (**e**) DHPG (100 μM, 5 min) evoked LTD in VTA DA neurons of adult mice ($p<0.001$, n=6, $t_{10}=19.38$), but not in adolescent mice ($p=0.10$, n=7, $t_{12}=1.76$).

The following figure supplements are available for figure 1:

**Figure supplement 1.** Identification of lateral VTA DA neurons in mouse midbrain slices.

**Figure supplement 2.** Adolescent mice are more susceptible than adult mice to cocaine-induced LTP in VTA DA neurons.

**Figure supplement 3.** VTA slices from adolescent mice more susceptible to cocaine-induced LTP in vitro.

**Figure supplement 4.** Basal p-eIF2α phosphorylation levels are similar in the VTA of adult and adolescent mice.

## Cocaine selectively reduces eIF2α phosphorylation in the VTA of adolescent mice

Protein synthesis is required for both cocaine-induced LTP (**Argilli et al., 2008**) and mGluR-LTD (**Mameli et al., 2007**) in VTA DA neurons, as well as cocaine-induced changes in behavior (**Sorg and Ulibarri, 1995**; **Kuo et al., 2007**). Given that translation rates in the brain decrease significantly with age (**Vargas and Castaneda, 1983**), we examined whether a low dose of cocaine is sufficient to trigger LTP in VTA DA neurons and induce CPP in adolescent mice (**Figure 1a** and **1c**) also activates a particular translational control program in the VTA of these mice. To this end, we measured the activity of key signaling pathways impinging on translation initiation (**Buffington et al., 2014**). We found that a low dose of cocaine (5 mg/kg) reduced the amount of phosphorylated eIF2α (p-eIF2α) only in VTA slices from adolescent mice (**Figure 2a** and **2b**). By contrast, a higher dose of cocaine

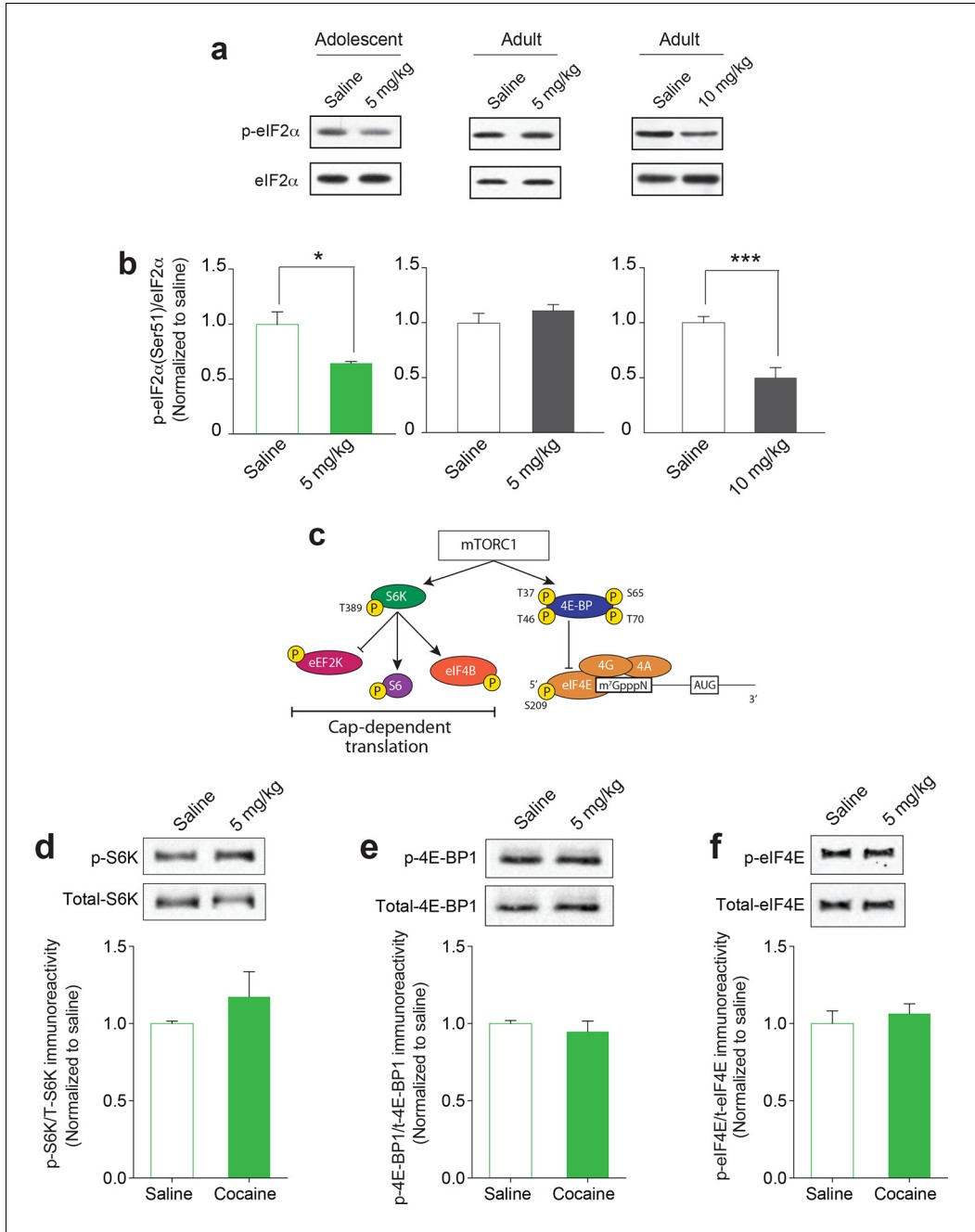

**Figure 2.** A low dose of cocaine selectively reduces p-eIF2$\alpha$ in the VTA of adolescent mice. (**a–b**) A low dose of cocaine (5 mg/kg) reduced p-eIF2$\alpha$ in the VTA of adolescent (p<0.05, n=5 per group, $t_8$=3.029), but not adult mice (p=0.329, n=3 per group, $t_4$=1.110). A higher dose of cocaine (10 mg/kg) was needed to reduce p-eIF2$\alpha$ in VTA of adult mice (p<0.001, n=6 per group, $t_{10}$=4.640). (**c**) Schematic of mTORC1- and eIF4E-mediated translation. In abilescnt mice, a low dose of cocaine (5 mg/kg) did not significantly alter phosphorylation of S6K at Thr-389 (**d**), 4E-BP1 at Thr-37 and Thr-46 (**e**) and eIF4E at Ser209 (**f**). Western blots are shown on top and quantification for each phospho-protein/total-protein is shown at the bottom (n=3/3 saline/cocaine; S6K, p=0.3467, $t_4$=01.066a; 4E-BP1, p=0.5031, $t_4$=0.7351; eIF4E, p=0.5669, $t_4$=0.6233). Plots are mean ± s.e.m.

The following figure supplement is available for figure 2:

**Figure supplement 1.** Doses of cocaine which lower p-eIF2$\alpha$ in the VTA have no effect in nucleus accumbens (NAc).

(10 mg/kg) was required to decrease p-eIF2α levels in VTA slices from adult mice (*Figure 2a* and *2b*). Importantly, a signle injection of cocaine failed to alter p-eIF2α levels in the nucleus accumbens (*Figure 2—figure supplement 1*), another brain region involved in addiction (*Kalivas and Volkow, 2011*). Moreover, the lack of effect on other translational signaling pathways in adolescent VTA neurons by the same low dose of cocaine (*Figure 2d–f*) highlights the selective involvement of p-eIF2α-mediated translational control during the period of heightened adolescent vulnerability to cocaine addiction. Thus, eIF2α is a newly identified effector of cocaine action.

## Reduced eIF2α phosphorylation-mediated translational control renders adult mice more susceptible to cocaine-evoked LTP and behavior

If reduced eIF2α phosphorylation enhances the susceptibility of adolescents to the effects of cocaine, then decreasing its phosphorylation in adults should increase their vulnerability. To test this idea, we injected a low dose of cocaine (5 mg/kg) into adult wild-type (WT) *Eif2s1$^{S/S}$* mice and *Eif2s1$^{S/A}$* heterozygous knock-in mice (where a single phosphorylation site at serine 51 is replaced by alanine) (*Scheuner et al., 2001*). In *Eif2s1$^{S/A}$* mutant mice, eIF2α phosphorylation was significantly reduced in the VTA (*Figure 3—figure supplement 1*). As predicted, a low dose of cocaine (5 mg/kg) induced LTP in VTA DA neurons from *Eif2s1$^{S/A}$* mutant mice but not in VTA DA neurons from WT littermates (*Figure 3a*). In addition, application of a low concentration of cocaine (1 μM) in vitro was sufficient to induce LTP in VTA DA neurons from *Eif2s1$^{S/A}$* mutant mice, but not in those from adult WT controls (*Figure 3—figure supplement 2*). Consistent with the LTP results, low doses of cocaine (5 mg/kg) induced CPP only in *Eif2s1$^{S/A}$* mutant mice (*Figure 3b*). Hence, like adolescent mice, adult mice with reduced p-eIF2α are more susceptible to cocaine-evoked LTP (both in vivo and in vitro) and drug-induced behavior.

We next asked whether acute pharmacological inhibition of p-eIF2α-mediated translation in adult mice renders animals more susceptible to a low dose of cocaine. Phosphorylation of eIF2α inhibits general protein synthesis rates by binding to and inhibiting the guanine nucleotide exchange factor (GEF) eIF2B that is required for eIF2 activation. We therefore used a recently-discovered small molecule inhibitor ISRIB (*Sidrauski et al., 2013*), which potently blocks p-eIF2α-mediated translational effects by promoting eIF2B activity (*Sekine et al., 2015*; *Sidrauski et al., 2015*). Adult WT mice injected with ISRIB (2.5 mg/kg) and a low dose of cocaine (5 mg/kg) showed both LTP in VTA (*Figure 3c*) and drug-induced CPP (*Figure 3d*). Note that neither cocaine nor ISRIB alone triggered LTP or CPP in adult WT mice (*Figure 3—figure supplement 3*). These results support the notion that reduced p-eIF2α–mediated translational control renders adult mice more susceptible to the synaptic and behavioral effects of cocaine.

Given that mGluR-LTD in the VTA of adolescent mice is impaired (*Figure 1e*) and adult mice with reduced p-eIF2α–mediated translational control resemble adolescent mice in their susceptibility to cocaine-induced synaptic potentiation and behavior (*Figure 1* and *3*), we predicted that mGluR-LTD might be deficient in adult mice with reduced p-eIF2α–mediated translational control. Consistent with this prediction, mGluR-LTD was impaired in VTA DA neurons from mice with reduced p-eIF2α (*Eif2s1$^{S/A}$* mice) and adult WT mice injected with ISRIB (*Figure 3e–f*). Thus, our parallel genetic and pharmacological experiments provide strong evidence that reducing p-eIF2α-mediated translational control in the VTA of adult mice makes them more like adolescents with respect to mGluR-LTD, and cocaine-evoked LTP and CPP.

## Adolescent mice with increased p-eIF2α in the VTA are more resistant to cocaine-evoked LTP and behavior

To examine whether increasing p-eIF2α in the VTA of adolescent mice is sufficient to confer resistance to low doses of cocaine, we administered Sal003, an inhibitor of eIF2α phosphatases (*Boyce et al., 2005*; *Robert et al., 2006*) (*Figure 4a* and *4b*), directly into the VTA of young mice to promote eIF2α phosphorylation locally (*Figure 4c* and *Figure 4—figure supplement 1*). As expected, a low dose of cocaine (5 mg/kg) induced LTP in adolescent mice locally infused with vehicle (*Figure 4b*), but not in VTA DA neurons from adolescent mice infused with Sal003 (*Figure 4b*). The Sal003-mediated increase in p-eIF2α also blocked the LTP evoked by cocaine in vitro in brain slices (*Figure 4d–e*), further supporting the in vivo experiments.

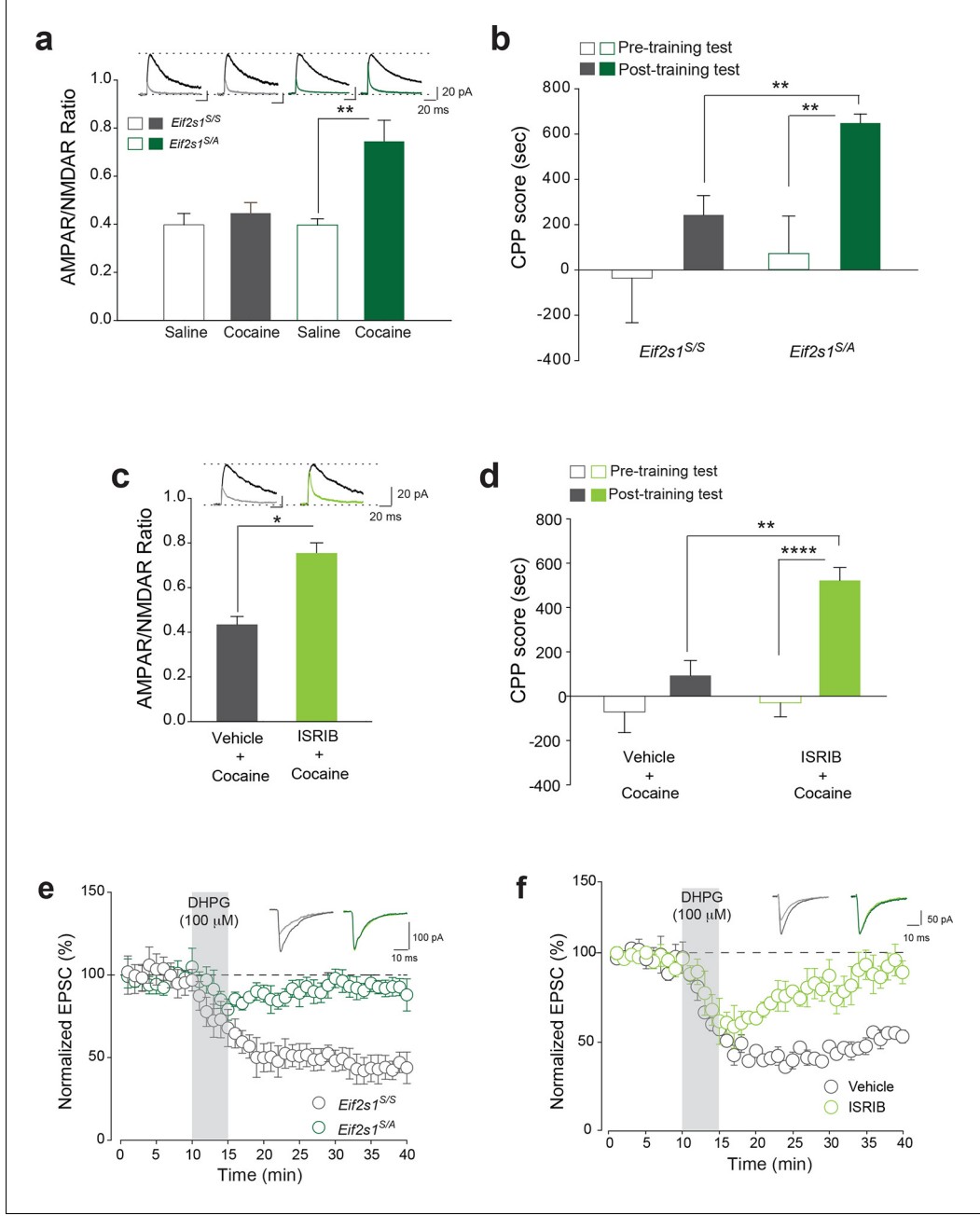

**Figure 3.** Decreasing p-eIF2α makes adult mice more susceptible to cocaine-induced LTP and behavior. (a–b) A low dose of cocaine (5 mg/kg) induced both LTP in VTA DA neurons (a, $p<0.05$, n=5, $t_8=4.193$) and CPP in adult $Eif2s1^{S/A}$ mice (b, $p<0.01$, n=7, $t_{12}=3.411$) compared to $Eif2s1^{S/S}$ mice (a, $p=0.89$, n=5, $t_8=0.14$; b, $p=0.2170$, n=7, $t_{12}=1.303$). (c–d) A low dose of cocaine (5 mg/kg) elicited LTP (c, $p<0.001$, n=6, $t_{10}=3.43$) and CPP (d, $p=0.1761$, n=8 vehicle+cocaine, $t_{14}=1.425$; $p<0.0001$, n=16 ISRIB+cocaine, $t_{30}=2.433$) in ISRIB-injected adult mice compared to vehicle-injected mice. (e–f) DHPG (100 μM, 5 min) induced LTD in WT adult VTA DA neurons (e, $p<0.001$, n=5, $t_8=20.3$) and vehicle-injected WT adult mice (f, $p<0.001$, n=5, $t_8=5.17$), but not in $Eif2s1^{S/A}$ mice (e, $p=0.26$, n=7, $t_{12}=1.2$) and ISRIB-injected mice (f, $p=0.42$, n=4, $t_6=0.86$).

The following figure supplements are available for figure 3:

**Figure supplement 1.** eIF2α phosphorylation is reduced in VTA from adult $Eif2s1^{S/A}$ mice.

**Figure supplement 2.** Decreasing p-eIF2α makes VTA slices from adult mice more susceptible to cocaine-induced LTP in vitro.

*Figure 3 continued on next page*

*Figure 3 continued*

**Figure supplement 3.** In adult mice, systemic administration of ISRIB alone failed to induce LTP in VTA DA neurons and CPP.

To determine whether changes in phosphorylation of eIF2α in the VTA and the behavioral susceptibility to cocaine are causally related, we assessed CPP in adolescent mice infused with vehicle or Sal003 directly into the VTA. Low doses of cocaine elicited CPP in vehicle-infused but not in Sal003-infused adolescent mice (*Figure 4f*). Moreover, as expected, Sal003 induced mGluR-LTD in VTA slices from adolescent mice (*Figure 4g*). Hence, a direct increase in p-eIF2α in the VTA of adolescent mice blocks the susceptibility to cocaine-induced LTP and behavior by promoting an opposing LTD in VTA DA neurons.

## OPHN1 knock-down in the VTA renders adult mice more susceptible to cocaine-evoked LTP and behavior

We next studied the mechanism by which reduced p-eIF2α-mediated translation renders adult mice more susceptible to the effects of cocaine. Given that *a)* eIF2α phosphorylation-mediated translational control is both necessary and sufficient for mGluR-LTD in the hippocampus (*Di Prisco et al., 2014*) and VTA (*Figures 3e, f* and *4g*), *b)* eIF2α phosphorylation blocks general translation but selectively triggers translation of a few select mRNAs during mGluR-LTD (including oligrophrenin-1 (*Ophn1*) mRNA) (*Di Prisco et al., 2014*), and *c)* translation of *Ophn1* mRNA is required for mGluR-LTD (*Di Prisco et al., 2014*; *Nadif Kasri et al., 2011*), we predicted that in adult mice with reduced OPHN1 levels in VTA, a low dose of cocaine (5 mg/kg) would induce LTP and CPP. As anticipated, in adult mice injected with a specific shRNA against *Ophn1* (*Ophn1*-shRNA) in the VTA, this low dose of cocaine (5 mg/kg) triggered LTP in the VTA in vivo (*Figure 5a*) and induced CPP (*Figure 5c*). However, the same low dose of cocaine failed to do so in mice injected with a control (scrambled) shRNA (*Figure 5a* and *5b*). Hence, like adolescent mice (*Figure 1a* and *1c*) or adult mice with reduced p-eIF2α–mediated translation (*Figure 3a* and *3b*), adult mice with reduced OPHN1 levels in the VTA are more sensitive to the effects of cocaine.

To assess the causal relationship between p-eIF2α and OPHN1 during cocaine-evoked LTP, we directly applied cocaine (5 µM) and Sal003 (20 µM) to brain slices in vitro and recorded LTP 3–5 hr after exposure, as previously described (*Argilli et al., 2008*). Remarkably, Sal003 blocked cocaine-induced LTP in control VTA DA neurons, but not in VTA DA neurons in which OPHN1 was reduced by *Ophn1*-shRNA (*Figure 5c*). In vivo injections of cocaine are known to induce LTP in the VTA by replacing postsynaptic AMPARs containing the GluR2 subunit with calcium-permeable AMPARs lacking the GluR2 subunit (*Bellone and Lüscher, 2006*). In order to investigate whether cocaine-induced LTP in vitro involves a similar process, we measured rectification (manifested as lower amplitude AMPAR EPSCs measured at positive holding potentials vs. those measured at negative potentials), a hallmark of GluR2-lacking AMPARs (*Liu and Zukin, 2007*). We recorded EPSCs at -70, 0 and +40 mV to calculate the rectification index and found that Sal003 blocked the increased inward rectification induced by in vitro application of cocaine in WT slices, but not in slices from OPHN1-deficient mice (*Figure 5d–g*). Collectively, these data indicate that OPHN1 is a specific target by which eIF2α phosphorylation regulates plasticity changes in VTA DA neurons.

## Different drugs of abuse decrease eIF2α phosphorylation in the VTA

Through their actions on distinct receptors, different drugs of abuse induce LTP in VTA DA neurons, thus reinforcing drug-seeking behavior (*Bowers et al., 2010*; *Lüscher and Malenka, 2011*). To test the effects of other addictive drugs on the phosphorylation of eIF2α in VTA, we treated mice with methamphetamine, nicotine, and ethanol at doses known to evoke LTP in VTA DA neurons (*Saal et al., 2003*). We found that, like cocaine, these drugs with very different mechanisms of action, all reduced p-eIF2α in VTA of adult mice (*Figure 6*).

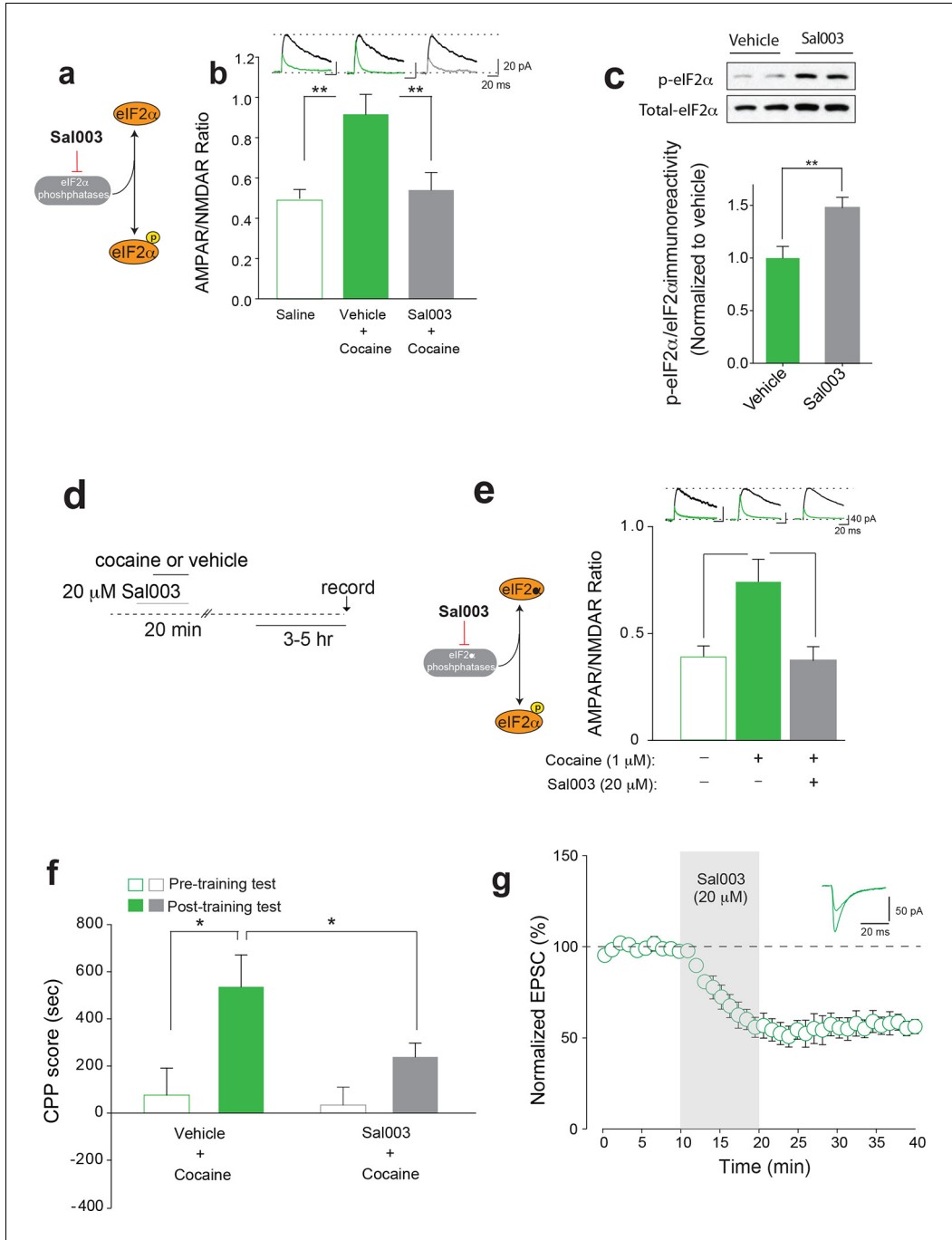

**Figure 4.** Increasing p-eIF2$\alpha$ in young mice blocks cocaine-induced LTP and behavior. (a) Schematic showing Sal003 mechanism of action. (b–c) Infusion of Sal003 into the VTA blocked cocaine-induced potentiation (c, p<0.001, n=5 per group, $t_8$=3.81) and increased p-eIF2$\alpha$ in the VTA (p<0.01, n=7/6 vehicle/Sal003, $t_{11}$=3.172). (d) Schematic of experimental design. (e) Direct application of cocaine (1 $\mu$M) induced LTP 3–5 hr post-treatment (p<0.05, n=11/6 vehicle/cocaine, $F_{2,20}$=7.48), whereas Sal003 prevented it (p<0.05, n=11/6, vehicle/cocaine+Sal003, $F_{2,20}$=7.48, cocaine vs. cocaine+Sal003). Representative traces of AMPAR and NMDAR EPSCs (top). (f) Infusion of Sal003 into the VTA blocked CPP (p<0.05, n=7 vehicle+cocaine, $t_{12}$=2.592; p=0.1147, n=10 Sal003+cocaine, $t_{18}$=1.892) in adolescent mice. (g) Application of Sal003 (20 $\mu$M, 10 min), a selective inhibitor of eIF2$\alpha$ phosphatases, induced LTD in VTA DA neurons from adolescent mice (p<0.001, n=6, $t_{10}$=9.517). Plots are mean ± s.e.m.

The following figure supplement is available for figure 4:

*Figure 4 continued on next page*

*Figure 4 continued*

**Figure supplement 1.** Sites of Sal003 infusions into VTA at seven rostrocaudal planes and corresponding increase in p-eIF2α.

## Discussion

Adolescence is a period of heightened susceptibility to drug addiction (*Chambers et al., 2003*; *Kandel et al., 1992*), but little is known about the underlying biological mechanisms. Changes in

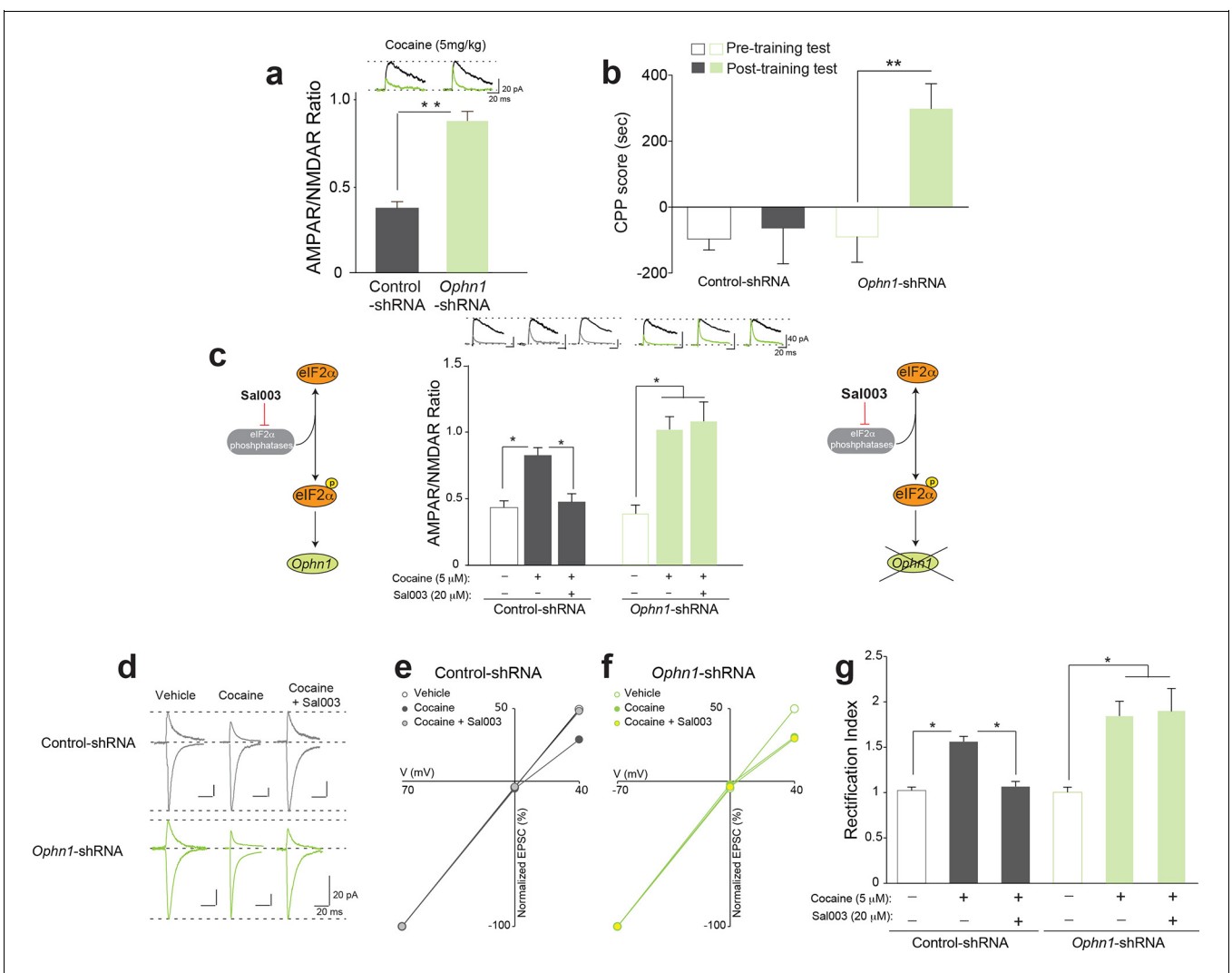

**Figure 5.** Decreasing OPHN1 levels in VTA DA neurons makes adult mice more susceptible to cocaine-induced LTP. (a) A low dose of cocaine (5 mg/kg) induced LTP in adult *Ophn1*-shRNA injected VTA DA neurons (a, Right, p<0.01, n=5, $t_8$=5.464); above representative traces of AMPAR and NMDAR EPSCs (top). (b) Low doses of cocaine (5 mg/kg) induced CPP in mice locally injected with *Ophn1*-shRNA (p<0.01, n=14, $t_{26}$=3.600), but not in control mice injected with scrambled shRNA (p=0.7829, n=4, $t_6$=0.2882). (c) Sal003 (20 µM) blocked the cocaine-induced LTP in the VTA of control shRNA-injected mice (p<0.01, n=6/6/7 vehicle/cocaine/cocaine+Sal003, $F_{2,16}$=13.03), but failed to do so in *Ophn1*-shRNA VTA DA neurons (p=0.29, n=6/6/11, vehicle/cocaine/cocaine+Sal003, $F_{2,20}$=4.29, cocaine vs. cocaine+Sal003; p<0.05 vehicle vs. cocaine or cocaine+Sal003). (d) Representative sample traces of AMPAR EPSCs. (e–f) *I-V* plots. (g) Cocaine increased the rectification index in control-shRNA injected VTA neurons while Sal003 blocked it (p<0.001, n=6/6/7 vehicle/cocaine/cocaine+Sal003, $F_{2,16}$=30.30, cocaine vs. vehicle or cocaine vs. cocaine+Sal003), whereas both cocaine and cocaine+Sal003 increased the rectification index in VTA DA neurons from *Ophn1*-shRNA-injected mice (p<0.05, n=6/6/11 vehicle/cocaine/cocaine+Sal003, $F_{2,20}$=3.92, vehicle vs. cocaine or cocaine+Sal003; p=0.80 cocaine vs. cocaine+Sal003). Plots are mean ± s.e.m.

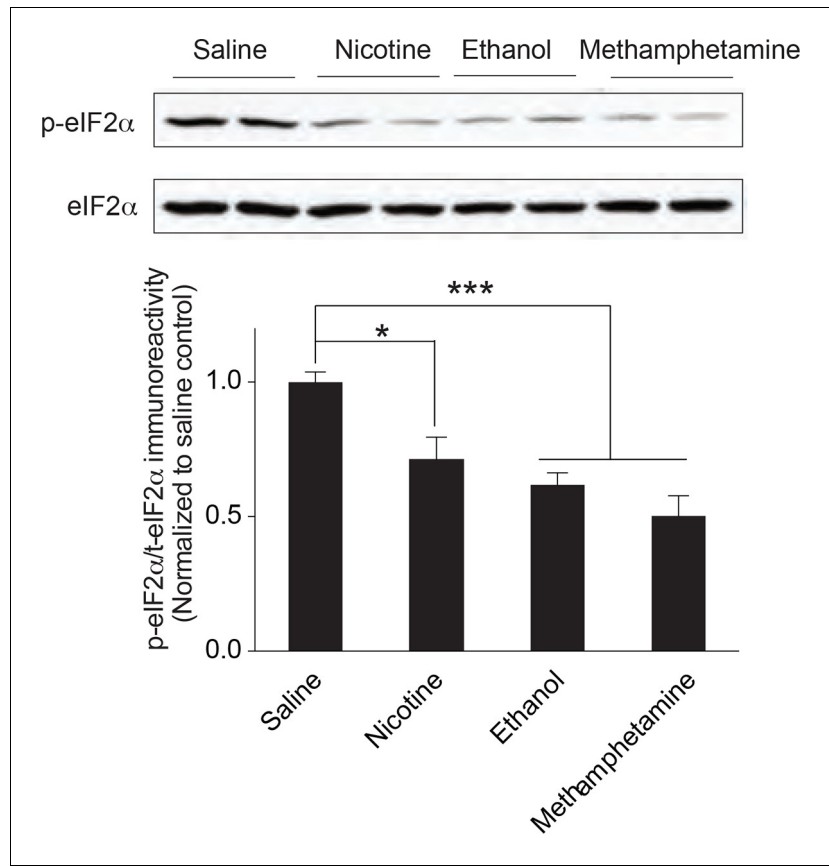

**Figure 6.** Multiple drugs of abuse reduce p-eIF2α in VTA of adult mice. (a) i.p. injection of nicotine (1 mg/kg), ethanol (2 g/kg), or methamphetamine (1 mg/kg) reduces p-eIF2α in VTA (n=5 per group; Saline vs. nicotine, $p<0.05$, $t_8=2.879$; ethanol, $p<0.001$, $t_8=6.278$ methamphetamine, $p<0.001$, $t_8=5.449$).

gene expression in key reward areas have been shown to play a critical role in drug-induced changes in synaptic potentiation and reward-related behavior (*Hyman et al., 2006*; *Robison and Nestler, 2011*). Until now, most of the research in this area has focused on how transcriptional control (*via* key transcription factors, such as △FosB and CREB) or epigenetic mechanisms (*via* histone acetylation and methylation, DNA methylation, and non-coding RNAs) contribute to addiction-related behavior (*Robison and Nestler, 2011*). Our focus on translational control was based on several key observations. Compared to the brains of adolescent mice—which are more vulnerable to drugs of abuse—protein synthesis is reduced in the brains of adult mice (*Vargas and Castaneda, 1983*; *Hovda et al., 2006*; *Sun et al., 1995*). Furthermore, translational control of gene expression is the ultimate step in the functional output of gene expression (*Sonenberg and Hinnebusch, 2009*) and neurons could regulate protein synthesis without altering mRNA synthesis or transport, allowing for local control of protein synthesis at synapses (*Holt and Schuman, 2013*). Moreover, protein synthesis is required for cocaine-evoked LTP in VTA DA neurons (*Argilli et al., 2008*; *Yuan et al., 2013*), as well as cocaine-induced behavior (*Sorg and Ulibarri, 1995*; *Kuo et al., 2007*).

Our findings reveal a critical role for p-eIF2α-mediated translational control in the heightened susceptibility of adolescent mice to the initial synaptic and behavioral adaptations induced by cocaine. In adolescent, but not in adult mice, a low dose of cocaine selectively lowers eIF2α phosphorylation in the VTA (*Figure 2a–b*), thereby eliciting LTP and drug-induced behavior. It is noteworthy that we could not detect a significant difference in the baseline phosphorylation of eIF2α between adolescent and adult mice (*Figure 1—figure supplement 4*). While the precise mechanism by which drugs of abuse lower phosphorylation of eIF2α in the VTA is currently under investigation, we hypothesize that drugs of abuse either inhibit the activity and/or expression of one of the eIF2α kinases or

stimulate the activity of eIF2α phosphatases to lower p-eIF2α levels in the VTA. We suspect that adolescents are more prone to either (or both) of these processes. Genetic reduction of eIF2α phosphorylation or pharmacologically blunting its effects with ISRIB in adult mice enhances susceptibility to cocaine, mimicking the increased vulnerability of adolescent animals (*Figure 3*). By contrast, a local increase in p-eIF2α in the VTA blocked cocaine-evoked LTP and cocaine-induced behavior in adolescent mice (*Figure 4*). These effects converge on p-eIF2α–mediated translation of OPHN1, which is selectively synthesized in neurons under conditions where eIF2α is phosphorylated (*Di Prisco et al., 2014*) and leads to endocytic down-regulation of post-synaptic AMPARs (*Nadif Kasri et al., 2011*). Collectively, our data indicate that reduced p-eIF2α-mediated translation of *Ophn1* mRNA accounts for the adolescent hypersensitivity to cocaine.

Is p-eIF2α-mediated mGluR-LTD in VTA a defense mechanism that limits the early synaptic adaptations required for the initiation of addiction? A delicate interplay between LTP and mGluR-LTD in VTA DA neurons is believed to modulate cocaine's synaptic and behavioral effects (*Loweth et al., 2013*; *Luscher, 2013*). Cocaine induces LTP at VTA DA synapses by inserting new AMPARs into the postsynaptic membrane (*Ungless et al., 2001*). This process is reversed either by pharmacological or synaptic induction of mGluR-LTD in VTA by removing AMPARs from the post-synaptic DA neurons (*Bellone and Lüscher, 2006*). Conversely, reduced mGluR-LTD in the midbrain is postulated to enhance vulnerability to drug addiction (*Lüscher and Huber, 2010*). Our data provide the first compelling mechanistic evidence that p-eIF2α-mediated translational control of OPHN1 synthesis is a key mechanism underlying the reversal of cocaine-evoked LTP by mGluR-LTD. Thus, our findings provide a unifying model that explains how the two opposing forms of plasticity (cocaine-induced LTP and mGluR-LTD) are regulated by a single translational control mechanism.

Given that blocking mGluR function in the VTA renders cocaine-evoked LTP more long-lasting (*Mameli et al., 2009*), it will be interesting to determine whether decreasing p-eIF2α also leads to a persistent LTP in the VTA. These drug-induced persistent changes on excitatory afferents onto dopamine neurons in the VTA are particularly relevant since they may represent the cellular processes driving the progression from recreational use to chronic drug seeking (*Chen et al., 2008*). In addition, eIF2α phosphorylation may also be required for subsequent synaptic adaptation in other mesolimbic reward areas, such as the NAc, in response to chronic exposure to cocaine or after withdrawal.

The identification of a single common downstream mechanism of action of different drugs of abuse has been challenging. It is therefore significant that drugs of abuse that act on distinct receptors (cocaine, methamphetamine, nicotine and ethanol), and induce LTP (*Bowers et al., 2010*; *Lüscher and Malenka, 2011*) in VTA DA neurons, all reduce p-eIF2α in the VTA (*Figure 6*). Taken together with our findings that reduced p-eIF2α-mediated translational control increases the susceptibility of adolescent mice to cocaine, these observations raise an intriguing possibility that polymorphisms in the eIF2α signaling pathway could be associated with drug use in humans. Indeed, in an accompanying paper (*Placzek et al., 2016*), we provide evidence that eIF2α phosphorylation also controls adolescent hypersensitivity to nicotine-evoked synaptic potentiation and identify a polymorphism in the promoter of the *Eif2s1* gene (encoding eIF2α) that is associated with changes in reward-related activity in human smokers, as measured by functional magnetic resonance imaging. Thus, since eIF2α phosphorylation is reduced by a variety of addictive drugs, agents that selectively alter eIF2α phosphorylation-mediated translational control in key reward areas in the brain could be useful for the treatment of a broad range of addictive behaviors.

## Materials and methods

### Mice

All experiments were conducted using male and female mice from the C57Bl/6 background. *Eif2s1^{S/A}* and *Eif2s1^{A/A}*;ftg mice were previously described (*Di Prisco et al., 2014*). Mice were kept on a 12h/12h light/dark cycle (lights on at 7:00 am) and had access to food and water *ad libitum*. During mid-adolescence (postnatal day 35–40) (*Spear, 2000*; *Laviola et al., 2003*), mice show characteristic behavior patterns, including impulsiveness and risk-taking (*Laviola et al., 2003*). We therefore selected five week-old mice (35–42 postnatal days) as adolescents, and 3–5 month old mice as

adults. Animal care and experimental procedures were approved by the institutional animal care and use committee (IACUC) at Baylor College of Medicine, according to NIH Guidelines.

No statistical methods were used to predetermine sample sizes. All sample sizes meet the criteria for corresponding statistical tests—our sample sizes are similar to those reported in previous publications (*Ungless et al., 2001*; *Saal et al., 2003*; *Bellone and Lüscher, 2006*; *Argilli et al., 2008*; *Koo et al., 2012*). For behavioral and biochemical studies, mice were arbitrarily assigned to control and treatment groups. These experiments were performed and analyzed blind to treatment conditions and/or genotype.

## Drug treatment

All drugs of abuse were dissolved in 0.9% saline and injected in a volume of 5 ml/kg. Cocaine hydrochloride, (−)-nicotine hydrogen tartrate, and USP-grade 95% ethanol were obtained from Sigma-Aldrich (St. Louis, MO). Racemic methamphetamine hydrochloride was a kind gift from Dr. Kristen Horner (Mercer University School of Medicine). ISRIB (P. Walter) was dissolved in DMSO and further diluted in PEG-400 (1:1 ratio) as previously described (*Di Prisco et al., 2014*). For both electrophysiological and behavioral experiments, ISRIB (2.5 mg/kg) or vehicle (DMSO/PEG-400, 2 ml/kg) was injected 90 min before cocaine or saline injection, respectively. Sal003 (Tocris Biosciences, R&D Systems, Minneapolis, MN) was dissolved in DMSO and further diluted in 0.9% saline. Sal003 (20 μM) or vehicle (0.4% DMSO in saline) was infused bilaterally into the VTA as summarized in the 'Cannulation and Sal003 infusion' section.

## Slice electrophysiology

Electrophysiological recordings were performed as previously described (*Ungless et al., 2001*; *Di Prisco et al., 2014*) and the investigators remained blind to genotypes. Each electrophysiological experiment was replicated at least three times. Briefly, mice were anesthetized with a mixture of ketamine (100 mg/kg), xylazine (10 mg/kg), and acepromazine (3 mg/kg). Mice were transcardially perfused with an ice-cold, oxygenated solution containing (in mM) NaCl, 120; NaHCO$_3$, 25; KCl, 3.3; NaH$_2$PO$_4$, 1.2; MgCl$_2$, 4; CaCl$_2$, 1; dextrose, 10; sucrose, 20. Horizontal slices (225–300 mm thick) containing the VTA were cut from the brains of adolescent (5 weeks old) or adult (3–5 months old) C57BL/6J mice with a vibrating tissue slicer (VF-100 Compresstome, Precisionary Instruments, San Jose, CA, or Leica VT 1000S, Leica Microsystems, Buffalo Grove, IL), incubated at 34°C for 40 min, kept at room temperature for at least 30 min prior to recording before they were transferred to a recording chamber where they were continuously perfused with artificial cerebrospinal fluid (ACSF) at 32°C and a flow rate of 2–3 ml/min. The recording ACSF was different from the cutting solution in the concentration of MgCl$_2$ (1 mM) and CaCl$_2$ (2 mM). Recording pipettes were made from thin-walled borosilicate glass (TW150F-4, WPI, Sarasota, FL). After filling with intracellular solution (in mM): 117 CsMeSO3; 0.4 EGTA; 20 HEPES; 2.8 NaCl, 2.5 ATP-Mg 2.0; 0.25 GTP-Na; 5 TEA-Cl, adjusted to pH 7.3 with CsOH and 290 mOsmol/l, they had a resistance of 3–5 MΩ. For studies of AMPAR current rectification, spermine (100 μM) was added to the internal solution, which blocks GluR2-lacking receptors at depolarized potentials.

Data were obtained with a MultiClamp 700B amplifier, digitized at 20 kHz with a Digidata 1440A, recorded by Clampex 10 and analyzed with Clampfit 10 software (Molecular Devices). Recordings were filtered online at 4 kHz with a Bessel low-pass filter. A 2 mV hyperpolarizing pulse was applied before each EPSC to evaluate the input and access resistance (Ra). Data were discarded when Ra was either unstable or greater than 25MΩ, holding current was >200 pA, input resistance dropped >20% during the recording, or EPSCs baseline changed by >10%. Traces illustrated in Figures are averages of 10–15 consecutive traces.

After establishing a gigaohm seal (>2GΩ) and recording stable spontaneous firing in cell-attached, voltage clamp mode (-70 mV holding potential), cell phenotype was determined by measuring the width of the cell-attached action potential (*Figure 1—figure supplement 1*). AMPAR/NMDAR ratios were calculated as previously described (*Ungless et al., 2001*). Briefly, neurons were voltage-clamped at +40 mV until the holding current stabilized (at <200 pA). Monosynaptic EPSCs were evoked at 0.05 Hz with a bipolar stimulating electrode placed 50–150 μm rostral to the lateral VTA. Picrotoxin (100 μM) was added to the recording ACSF to block GABA$_A$R-mediated IPSCs. After recording the dual-component EPSC, DL-AP5 (100 μM) was bath-applied for 10 min to remove the

NMDAR component, which was then obtained by offline subtraction of the remaining AMPAR component from the original EPSC. The peak amplitudes of the isolated components were used to calculate the AMPAR/NMDAR ratios. Rectification indices were calculated as the ratio of the chord conductance of evoked EPSCs at a negative holding potential (-70 mV) to the chord conductance at a positive holding potential (+40 mV) obtained in the presence of 100 μM DL-AP5, as previously described (*Bellone et al., 2011*). Picrotoxin and DL-AP5 were purchased from Tocris Bioscience and all other reagents and experimental compounds were obtained from Sigma-Aldrich.

Experiments applying drugs in vitro were performed as previously described (*Argilli et al., 2008*) with slight modifications. Briefly, slices were incubated with cocaine (1 μM or 5 μM) and Sal003 (20 μM) for 15 and 20 min, respectively, as shown in *Figure 3B*. After treatment, slices were transferred (twice) to a 35 mm polycarbonate dish containing regular ACSF for complete drug washout and allowed to recover for 2–4 hr. Whole-cell recordings were then conducted 3–5 hr after the end of drug exposure.

## Virus Injection

AAV5-Cre (Titer: 1.0e13GC/ml) was purchased from Vector Biolabs (Cat#7012, Philadelphia, PA); Lentiviral constructs expressing *Ophn1* shRNA and scrambled shRNA were generously provided by Dr. Linda van Aelst (*Nadif Kasri et al., 2011*) (Cold Spring Harbor Laboratory) and viruses were produced by Gene Vector Core Laboratory (Baylor College of Medicine). Viral injections were performed as previously described (*Di Prisco et al., 2014*). Briefly, mice were anaesthetized with isoflurane (2–3%) and viruses (1–2 μl/site) were injected bilaterally at the rate of 0.1 μl/min, and an additional 10 min to allow for diffusion of viral particles. Injection coordinates, targeting VTA, were as follows (with reference to bregma): -2.50 AP, ± 0.45 ML, −4.50 DV. The incision was sutured after injection and mice were returned to home cages. Mouse body weight and signs of illness were monitored until full recovery from surgery (~1 week). Drug treatment and experiments were all performed at least three weeks after viral injection.

## Conditioned Place Preference (CPP)

The investigators were blind to the genotypes for the behavioral tests. CPP, performed as previously described (*Koo et al., 2012*), was assessed over 6 days using an unbiased procedure and a standard two-chamber CPP apparatus (Ugo Basile, Varese, Italy). Animal behavior was videotaped with an overhead camera and analyzed by ANY-maze software (Stoelting, Wood Dale, IL). The difference in the time spent in cocaine-paired side versus saline-paired side was calculated as the CPP score. On day 1, a mouse was placed in the chamber with the doors removed for a 30 min pre-training test and the baseline preference was calculated. Mice with strong pre-training preference to any chamber (CPP score >540 s) were excluded from the experiment (<10% of all mice tested). On the following four days, training sessions were performed once a day. On alternate days, mice were given injections of cocaine (5 mg/kg or 10 mg/kg, i.p.) or 0.9% saline (5 ml/kg, i.p.) immediately before being confined to the cocaine-paired or saline-paired chamber for 30 min and then returned to their home cages. On day 6, a test session identical to the pre-training test was conducted to determine the CPP scores.

## Cannulation and Sal003 infusion

Mice were anesthetized with isoflurane (2–3%) and mounted on a stereotaxic frame. Cannulae (26 gauge) were implanted bilaterally to target the VTA region at an angle of 15° from the midline at these coordinates: -3.16 mm AP, ± 0.63 mm ML, -3.72 mm DV (as determined from the Paxinos & Franklin atlas). Two jewelry screws were inserted into the skull and the cannulae were held in place by acrylic cement. A 33 gauge dummy probe was inserted into the guide to prevent clogging by tissue debris. Bilateral infusions [0.5 μl of vehicle (0.4% DMSO in saline) or Sal003 (20 μM)] were made *via* the implanted cannulae in freely-moving mice 30 min before cocaine or saline injection, driven by a motorized syringe pump (KdScientific) at the rate of 0.1 μl/min. After 5 min of infusion, the injector remained in the cannulae for an additional minute to allow diffusion of the solution. Cannula placements were visually confirmed in subsequent brain sections. For electrophysiological and behavioral experiments, mice were killed after all tests for histological confirmation. Brains were fixed in 4% paraformaldehyde and 80 μm sections cut for Nissl-staining to identify cannula placement. Only

mice with correct bilateral placements were included in the analyses. Cannulae and infusion accessories were custom-made by Plastics One (Roanoke, VA).

## Western blotting

VTA samples were micro-dissected from 1 mm coronal sections obtained using an acrylic mouse brain matrix (Stoelting). Briefly, mice were killed by an overdose of isoflurane and their brains were quickly removed and placed in ice-cold PBS-immersed brain matrix for sectioning. The section containing most of the VTA (typically 8th from rostral to caudal) was then transferred for microdissection of the VTA using scalpels over an ice-cold petri-dish. Samples were collected in pre-chilled microcentrifuge tubes and lysed in homogenizing buffer [200 mM HEPES, 50 mM NaCl, 10% Glycerol, 1% Triton X-100, 1 mM EDTA, 50 mM NaF, 2 mM $Na_3VO_4$, 25 mM β-glycerophosphate, and EDTA-free complete ULTRA tablets (Roche, Indianapolis, IN)]. Western blotting was performed as previously described (*Huang et al., 2013*). Primary antibodies for Western blotting were rabbit anti-p-eIF2α (Ser51) (1:1000, Cell Signaling Technology Laboratories, Danver, MA), mouse anti-total eIF2α (1:1000, Cell Signaling Technology Laboratories, Danver, MA), and mouse anti-β-actin (1:10,000, EMD Millipore, Billerica, MA).

## Statistical Analyses

All data are presented as mean ± s.e.m. Statistical analyses were performed using SigmaPlot (Systat Software). Data distribution normality and homogeneity of variance were assessed using the Shapiro-Wilk and Levene tests, respectively. The statistics were based on the two-sided Student's t test, or one- or two-way ANOVA with Tukey's HSD (or HSD for unequal sample sizes where appropriate) to correct for multiple *post hoc* comparisons. Within-groups variation is indicated by standard errors of the mean of each distribution, which are depicted in the graphs as error bars. $P<0.05$ was considered significant (*$P<0.05$, **$P<0.01$, ***$P<0.001$, ****$P<0.0001$).

## Acknowledgements

We thank Hongyi Zhou for assisting in the maintenance of mouse colony and members of the Costa-Mattioli laboratory for comments on the paper. We thank Gabriel Malek for his excellent assistance with optimizing the CPP experiments during his internship in the lab. This work was supported by grants from the National Institutes of Health to MCM (NIMH 096816, NINDS 076708) and JD (NIDA DA09411 and NINDS NS21229). PW is an Investigator of the Howard Hughes Medical Institute.

## Additional information

### Funding

| Funder | Author |
| --- | --- |
| National Institute of Mental Health | Mauro Costa-Mattioli |
| National Institute of Neurological Disorders and Stroke | John A Dani<br>Mauro Costa-Mattioli |
| Howard Hughes Medical Institute | Peter Walter |
| National Institute on Drug Abuse | John A Dani |

The funders had no role in study design, data collection and interpretation, or the decision to submit the work for publication.

### Author contributions

WH, Conducted behavioral and molecular experiments and analyzed data, Conception and design, Acquisition of data, Analysis and interpretation of data; ANP, Conducted electrophysiological experiments and analyzed data, Conception and design, Acquisition of data, Analysis and

interpretation of data; GVDP, Conducted electrophysiological experiments and analyzed data, Acquisition of data, Analysis and interpretation of data; SK, Conducted electrophysiology experiments and analyzed data, Conception and design, Acquisition of data, Analysis and interpretation of data; CS, Contributed to discussion of the project, Drafting or revising the article; KK, PW, Contributed to in-depth discussion of the project and editing of the manuscript, Drafting or revising the article; JAD, Contributed to in-depth discussion of the project, Drafting or revising the article; MCM, Designed the experiments, analyzed data and wrote the manuscript., Conception and design, Analysis and interpretation of data, Drafting or revising the article

## Author ORCIDs

Mauro Costa-Mattioli, (iD) http://orcid.org/0000-0002-9809-4732

## Ethics

Animal experimentation: Animal care and experimental procedures were approved by the institutional animal care and use committee (IACUC) at Baylor College of Medicine (Protocol number: AN-5068), according to NIH Guidelines.

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
