## [Decision Letter]

Thank you for submitting your work entitled "Translational control by eIF2α phosphorylation regulates vulnerability to the synaptic and behavioral effects of cocaine" for consideration by *eLife*. Your article has been reviewed by two peer reviewers, and the evaluation has been overseen by a Reviewing Editor and a Senior Editor.

The reviewers have discussed the reviews with one another and the Reviewing Editor has drafted this decision to help you prepare a revised submission

The reviewers like your work and although they thought that some additional experiments may strengthen the study, as per *eLife* policy, they decided that your study should be acceptable without further experiments. We would ask if some parts that are not addressed in this study (like the existence of additional unspecified mechanism that allows low dose cocaine to decrease levels of p-eIF2α in adolescents but not adults) could be discussed in the final version. The full comments of both reviewers are posted below for your general guidance in preparing a final version of the paper.

*Reviewer #1:*

This very well written manuscript reports novel animal model studies and highly mechanistic evidence that identifies eIF2α as a major player in cocaine's effects in the ventral tegmental area, and more importantly, identifies this molecule and translation control as a key reason for why adolescents are especially sensitive to cocaine. Importantly, the authors go well beyond implicating eIF2α-dependent translation, and go on to identify a likely target (OPHN1), the cellular mechanisms involved (in vivo and slice synaptic plasticity studies of LTP/LTD), the key brain region involved (VTA) and compelling behavioral data with a number of impressive rescuing experiments that turn the cocaine susceptibility from adults to adolescents and vice versa. This is by far one of the very best papers I have read this year, and the implications of these results are considerable, since the findings seem to apply to not only cocaine but also to other abused substances (methamphetamine, nicotine, and ethanol). Impressively, in a companion paper the authors report evidence (allelic variability) that this mechanism may be at play in human populations.

Reviewer #2:

This is a potentially interesting manuscript, from an excellent cadre of investigators, that ultimately suggests that adults may be less susceptible to addiction because adults don't respond as much to low doses of cocaine. In general, the manuscript attempts to demonstrate Eif2a's role in addiction, but makes a stronger case for its role in gating acute drug-induced VTA DA plasticity. This manuscript has bits of important data, but needs more experiments in order to become a seamless, outstanding story.

Comments:

Are baseline levels of p-eIF2α different in adolescent compared to adult mice? is the baseline ratio of p-eIF2α/total eIF2α different in adolescent compared to adult mice? If the authors wish to argue that low abundance of p-eIF2α is a permissive factor for induction of cocaine plasticity, and that this is a critical factor in the lower threshold for cocaine plasticity in adolescents, this seems like a critical experiment.

The authors make a case that increased translation of OPHN1 downstream of increased levels of p-eIF2α is what supports mGluR-LTD in adults and blocks/counteracts cocaine LTP in VTA DA neurons. Are baseline levels of OPHN1 lower in adolescent compared to adult mice? If one overexpresses OPHN1 in adolescent VTA DA neurons, would you get effective mGluR-LTD and block the ability of low dose cocaine to induce LTP/CPP?

Bottom line: the link between Eif2α and mGluR LTD would benefit from more work, beyond simply referencing previous publications.

There also seems to be some circular logic here – the authors have a lot of experiments to show that preexisting lower levels of p-eIF2α (or OPHN1) are the permissive factor that allows low dose cocaine to induce LTP / CPP, and the reason that they offer for why adolescents are more susceptible to drugs is that low dose cocaine can lead to reductions in p-eIF2α in adolescent but not adult VTA. Therefore, there must be some additional unspecified mechanism that allows low dose cocaine to decrease levels of p-eIF2α in adolescents but not adults.

What is it that determines why low dose of cocaine (or nicotine, or methamphetamine, or ethanol) can change the p-eIF2α/total eIF2α ratio in adolescent but not adult mice? In the absence of data for the above two bullet points (showing that there are preexisting lower levels of p-eIF2α / OPHN1 in adolescent mice), it would seem that this additional unspecified factor is what renders adolescents more vulnerable to cocaine plasticity, not translational control by p-eIF2α.

---

## [Author Response]

In response to the editor’s request, we have addressed the points raised by reviewer #2 in the Discussion section of the revised manuscript. In addition, we performed some additional experiments and the new data are now included in Figure 1—figure supplement 4.

1) We now provide evidence that basal eIF2α phosphorylation levels are similar in the VTA of adolescent and adult mice. We now include these data in Figure 1—figure supplement 4.

2) In the Discussion section (second paragraph) we speculate on the potential mechanism that could explain why a low dose of cocaine is able to reduce eIF2α phosphorylation in the VTA of adolescent mice. More specifically, we propose that cocaine (and other drugs of abuse) either inhibit the activity of a given eIF2α kinase or promote the activity of eIF2α phosphatases, processes to which we suspect adolescents are more prone.

3)Reviewer# 2 indicates that Eif2α and mGluR LTD would benefit from more work, beyond simply referencing previous publications.While we based our study on our previous work (Di Prisco et al., Nat Neurosci, 2014), we would like to emphasize that here we provide gain- and loss-of-function evidence that eIF2α-mediated translational control is both required and sufficient for mGluR-LTD in the VTA (Figure 3 and Figure 4).